# Systemic inflammatory response index is associated with acute kidney injury following cardiac surgery: A retrospective cohort study using the MIMIC database

Huiliang Xie[1☉], Keyang Zheng[1☉], Kun Wang[1], Zixu Zhao[2], Rite Si[3,4], Jingyi Xiao[5], Yuzhe Yin[6], Xiayang Zhu[1]*

1 Department of General Practice, Beijing Nuclear Industry Hospital, Beijing, China, 2 Beijing Anzhen hospital, Capital medical school, Beijing, China, 3 Beijing Key Laboratory of Mental Disorders, National Clinical Research Center for Mental Disorders &National Center for Mental Disorders, Beijing Anding Hospital, Capital Medical University, Beijing, China, 4 Advanced Innovation Center for Human Brain Protection, Capital Medical University, Beijing, China, 5 The Sixth Clinical Medical School, Capital Medical University, Beijing, China, 6 Department of Cardiology, Fuwai Hospital, National Center for Cardiovascular Disease, Chinese Academy of Medical Science and Peking Union Medical College, Beijing, China

☉ These authors contributed equally to this work.
* xiayangzhuss@126.com

## Abstract

### Purpose

This study aimed to investigate the association between the Systemic Inflammatory Response Index (SIRI) and acute kidney injury (AKI) following cardiac surgery using the Medical Information Mart for Intensive Care (MIMIC) database, and to evaluate whether SIRI could serve as a potential risk marker associated with post-cardiac surgery AKI.

### Methods

We conducted a retrospective cohort study of 2,884 cardiac surgery patients from the MIMIC-IV database. SIRI was calculated as (neutrophil count × monocyte count)/ lymphocyte count. The primary outcome was AKI occurrence within seven days post-surgery. Logistic regression models and restricted cubic spline (RCS) analysis were used to assess the association between SIRI and AKI risk. Subgroup analyses were performed to evaluate potential effect modifiers.

### Results

Higher SIRI levels were significantly associated with increased AKI risk, even after adjusting for potential confounders (OR for highest vs. lowest quartile: 1.35, 95% CI: 1.04–1.77). A dose-response relationship was observed between SIRI and AKI severity (P for trend < 0.001). The association between SIRI and AKI risk was more

**Data availability statement:** The datasets analyzed for this study can be found in the Medical Information Mart for Intensive Care (MIMIC-IV) database(https://mimic.mit.edu/).

**Funding:** The author(s) received no specific funding for this work.

**Competing interests:** The authors have declared that no competing interests exist.

pronounced in patients with a history of myocardial infarction (OR: 1.261, 95% CI: 1.084–1.467) and those not using loop diuretics (OR: 2.306, 95% CI: 1.200–4.434).

## Conclusion

SIRI showed a modest but significant association with AKI following cardiac surgery. Its integration of multiple inflammatory cell types provided a comprehensive assessment of inflammatory status. The varying strength of association across different patient subgroups suggested the need for individualized risk assessment strategies. Further research is warranted to validate these findings and explore the underlying mechanisms.

## 1. Introduction

Cardiac surgery is an effective treatment for various heart diseases. However, the high incidence of postoperative acute kidney injury (AKI) remains a significant challenge for clinicians. Studies have shown that the incidence of AKI following cardiac surgery can be as high as 30–40% [1,2]. The occurrence of AKI not only prolongs hospital stays and increases medical costs but also significantly elevates both short-term and long-term mortality risks for patients [3,4]. Research indicates that AKI is independently associated with in-hospital mortality and increases the risk of progression to end-stage renal disease and major adverse cardiovascular events [4,5]. These data underscore the importance of AKI prevention and early identification.

In recent years, researchers have been dedicated to identifying factors that can accurately predict AKI following cardiac surgery. Traditional predictive factors include age, perioperative estimated glomerular filtration rate (eGFR), history of coronary heart disease (CHD) or chronic kidney disease (CKD), and preoperative use of calcium channel blockers or proton pump inhibitors [6]. Moreover, biomarkers such as N-terminal pro-brain natriuretic peptide have demonstrated strong predictive potential [1]. As our understanding of AKI deepens, an increasing body of evidence suggests that inflammation plays a crucial role in post-cardiac surgery AKI. Elevated inflammatory markers such as C-reactive protein (CRP) and neutrophil-to-lymphocyte ratio (NLR) have been associated with increased AKI risk following coronary artery bypass grafting and percutaneous coronary intervention (PCI), respectively [7,8]. Other cytokines, including interferon-γ, interleukin-16, and macrophage inflammatory protein-1α (MIP-1α), have been identified as potential prognostic biomarkers [9]. Despite these findings providing new insights into AKI prediction, current indicators still face challenges of insufficient specificity or limited predictive capability.

Among various inflammatory indicators, the Systemic Inflammatory Response Index (SIRI) has garnered attention due to its ability to comprehensively reflect the body's inflammatory state. SIRI was initially used to predict survival rates in pancreatic cancer patients undergoing chemotherapy, potentially aiding clinicians in improving treatment outcomes by identifying candidates for aggressive therapy [10]. Recent studies have demonstrated SIRI's value across multiple domains. In patients with

acute coronary syndrome, SIRI has emerged as an independent predictor of major adverse cardiovascular events, exhibiting optimal predictive performance [11]. In non-dialysis CKD patients, SIRI is independently associated with advanced CKD and can predict CKD progression [12]. Furthermore, a large-scale cohort study of US adults found that higher SIRI levels were associated with increased risks of all-cause and cardiovascular mortality [13]. More recently, SIRI has shown promise in predicting AKI in other clinical settings, such as sepsis and contrast exposure [14]. Although SIRI has demonstrated its utility in numerous fields, its association with post-cardiac surgery AKI has remained largely unexplored and warrants further investigation.

The primary objective of this study was to investigate the association between the SIRI index and AKI following cardiac surgery, and to evaluate whether SIRI could serve as a potential risk marker associated with AKI.

## 2. Methods

### 2.1 Study design and data source

This study employed a retrospective cohort design, utilizing the Medical Information Mart for Intensive Care (MIMIC-IV) database for analysis. The MIMIC database is a large, publicly available dataset containing detailed information on intensive care unit (ICU) patients, including hospitalization details, laboratory test results, and clinical diagnoses. Our research focused on patients who underwent cardiac surgery, aiming to explore the relationship between the SIRI and the occurrence of AKI following cardiac procedures. The data were accessed for research purposes in June 2024. All data used were fully de-identified, and authors had no access to identifiable personal information at any stage of data handling or analysis. Therefore, this study was exempt from ethical review. We strictly adhered to the ethical principles outlined in the Declaration of Helsinki.

### 2.2 Study population

The initial sample comprised all patients in the MIMIC database who underwent cardiac surgery. We identified eligible patients using cardiac surgery-related procedure codes from the ICD-9 and ICD-10 (codes are listed in Supplementary Material S1). Strict inclusion and exclusion criteria were applied to determine the final study cohort, as illustrated in Fig 1. Inclusion criteria were: (1) patients who underwent cardiac surgery (N = 76,943); (2) data from only the first admission

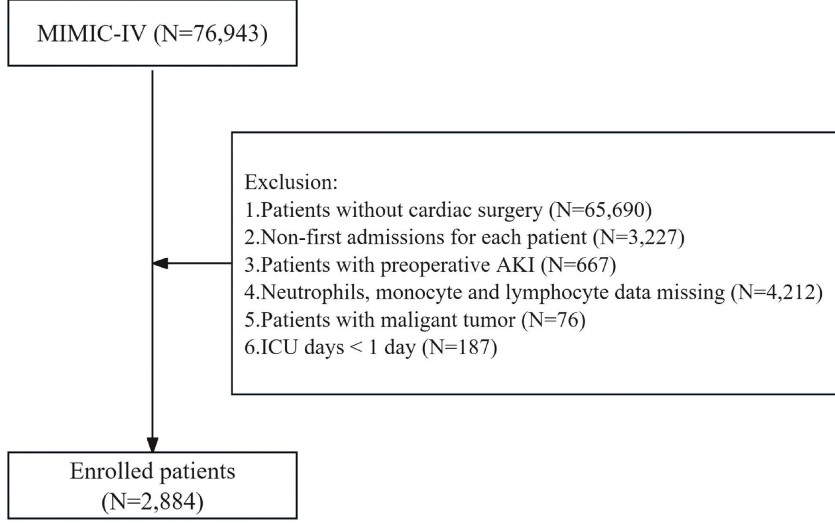

**Fig 1. Flow Chart of Patient Selection Process.**

or first ICU stay for each patient (N = 3,227). Exclusion criteria included: (1) patients with pre-existing AKI (N = 667); (2) patients with insufficient data to calculate SIRI (N = 4,212); (3) patients with malignant tumors (N = 76); (4) patients with ICU stays less than 1 day (N = 187). After screening, 2,884 patients were included in the final analysis. Cardiac surgical procedures included coronary artery bypass grafting (CABG), mitral valve repair, aortic valve replacement, mitral valve replacement, thoracic aorta replacement, tricuspid valve replacement, tricuspid valve repair, and aortic valve repair. For analysis purposes, surgical procedures were categorized into CABG and others (non-CABG). All clinical and laboratory data for included patients were derived from the first examination results after admission.

## 2.3  Variable definitions

The primary outcome variable was AKI, defined according to the Kidney Disease: Improving Global Outcomes (KDIGO) guidelines within seven days post-surgery [15]. The main exposure factor was SIRI, calculated as: (neutrophil count × monocyte count)/ lymphocyte count [10]. SIRI was categorized into four groups (Q1, Q2, Q3, Q4) based on quartiles for categorical analysis. All covariates were extracted from the MIMIC database, including demographic characteristics, laboratory indicators, medical history, and medication use.

## 2.4  Statistical analysis

Statistical analyses were performed using R software. Baseline characteristics of the study population were described using descriptive statistics. The normality of continuous variables was assessed using the Shapiro-Wilk test. Non-normally distributed continuous variables were described using median and interquartile range (IQR) and compared between groups using the Kruskal-Wallis test. Categorical variables were presented as frequencies and percentages and compared using the Chi-square test.

To investigate the association between SIRI and AKI risk, we constructed four logistic regression models: Model 1 was unadjusted; Model 2 adjusted for sex, age, and race; Model 3 further adjusted for hemoglobin (Hb), platelet count (PLT), serum creatinine (Scr), and eGFR; Model 4 additionally adjusted for history of hypertension, diabetes, heart failure, use of angiotensin-converting enzyme inhibitors (ACEIs) or angiotensin receptor blockers (ARBs), use of loop diuretics, and surgery type (CABG vs. others). Results were presented as odds ratios (OR), 95% confidence intervals (CI), and P-values.

We calculated the mean and standard deviation of SIRI for patients with no AKI, AKI stage 1, stage 2, and stage 3, and performed a P for trend test to evaluate the relationship between SIRI levels and AKI severity. To assess potential non-linear relationships between SIRI and AKI risk, we used restricted cubic spline (RCS) curves. Additionally, we conducted extensive subgroup analyses based on covariates and other AKI-related factors. We also performed interaction analyses to evaluate the potential influence of these factors on the relationship between SIRI and AKI risk. All statistical tests were two-sided, with P < 0.05 considered statistically significant.

## 3.  Results

### 3.1  Baseline characteristics

This study included 2,884 patients, comprising 2,154 males (74.69%) and 730 females (25.31%). The median age was 67 years (IQR: 60–74 years), and the median body mass index (BMI) was 28.40 kg/m$^2$ (IQR: 25.24–32.45 kg/m$^2$).

Table 1 presented the baseline characteristics of the study population, stratified into quartiles based on SIRI values. The median SIRI values and IQRs for each quartile were: Q1: 0.68 (0.44–0.89), Q2: 1.58 (1.34–1.82), Q3: 2.91 (2.49–3.40), and Q4: 6.12 (4.83–8.76). The incidence of AKI increased across SIRI quartiles, ranging from 74.20% in Q1 to 80.17% in Q4 (P = 0.038). Our findings revealed statistically significant trends in almost all baseline indicators across SIRI quartiles (most P-values <0.001). Individuals in higher SIRI quartiles were more likely to be male (81.55% vs 64.77%, P < 0.001) and white (75.45% vs 66.16%, P < 0.001). Compared to the lowest quartile (Q1), patients in the highest quartile

**Table 1. Baseline characteristics of study participants stratified by SIRI quartiles.**

| Variables | SIRI | | | | P-value |
|---|---|---|---|---|---|
| | Q1 | Q2 | Q3 | Q4 | |
| N | 721 | 721 | 721 | 721 | |
| Age(years), Median (IQR) | 67.0 (60.0-74.0) | 67.0 (60.0-74.0) | 68.0 (59.0-74.0) | 67.0 (59.0-74.0) | 0.641 |
| Sex | | | | | <0.001 |
| Male, n(%) | 467 (64.8%) | 531 (73.7%) | 568 (78.8%) | 588 (81.6%) | |
| Female, n(%) | 254 (35.2%) | 190 (26.4%) | 153 (21.2%) | 133 (18.5%) | |
| Race | | | | | <0.001 |
| White, n(%) | 477 (66.2%) | 515 (71.4%) | 536 (74.3%) | 544 (75.5%) | |
| Black/Asia/Hispanic, n(%) | 105 (14.6%) | 64 (8.9%) | 52 (7.2%) | 50 (7.0%) | |
| Other, n(%) | 139 (19.3%) | 142 (19.7%) | 133 (18.5%) | 127 (17.6%) | |
| BMI (kg/m$^2$), Median (IQR) | 27.3 (24.5-31.3) | 28.7 (25.6-32.3) | 28.6 (25.4-32.8) | 29.0 (25.4-33.4) | <0.001 |
| Systolic pressure (SBP) (mm/Hg), Median (IQR) | 112.0 (101.0-124.0) | 114.0 (101.0-124.0) | 112.0 (101.0-124.0) | 111.0 (101.0-123.0) | 0.611 |
| Diastolic pressure (DBP) (mm/Hg), Median (IQR) | 58.0 (52.0-65.0) | 59.0 (52.0-66.0) | 58.0 (52.0-65.0) | 59.0 (52.0-66.0) | 0.578 |
| HR(BPM), Median (IQR) | 80.0 (70.0-82.0) | 80.0 (72.0-83.0) | 80.0 (74.0-85.0) | 80.0 (75.0-88.0) | <0.001 |
| RR(RPM), Median (IQR) | 16.0 (14.0-17.0) | 16.0 (14.0-18.0) | 16.0 (14.0-18.0) | 16.0 (15.0-19.0) | <0.001 |
| Temperature (°C), Median (IQR) | 36.4 (35.8-36.7) | 36.4 (36.1-36.7) | 36.4 (36.1-36.8) | 36.6 (36.3-36.9) | <0.001 |
| RBC (10$^{12}$/L), Median (IQR) | 3.4 (2.9-4.2) | 3.6 (3.0-4.3) | 3.5 (3.0-4.2) | 3.6 (3.1-4.3) | <0.001 |
| Hb(g/L), Median (IQR) | 10.2 (8.7-12.4) | 10.7 (9.2-12.8) | 10.7 (9.1-12.7) | 10.6 (9.1-12.8) | <0.001 |
| RDW (%), Median (IQR) | 13.2 (12.7-14.0) | 13.2 (12.7-14.1) | 13.2 (12.6-14.1) | 13.6 (12.9-14.8) | <0.001 |
| WBC (10$^9$/L), Median (IQR) | 7.6 (6.0-10.2) | 9.1 (7.1-11.7) | 10.4 (7.8-13.9) | 11.9 (8.3-16.5) | <0.001 |
| PLT (10$^9$/L), Median (IQR) | 152.0 (118.0-194.0) | 164.0 (127.0-213.0) | 164.0 (129.0-209.0) | 180.0 (140.0-231.0) | <0.001 |
| Neutrophils(10$^9$/L), Median (IQR) | 6.0 (4.2-7.6) | 8.1 (6.6-9.9) | 10.6 (8.5-12.5) | 13.3 (10.6-16.5) | <0.001 |
| Lymphocyte(10$^9$/L), Median (IQR) | 2.0 (1.5-2.7) | 2.1 (1.6-2.7) | 2.0 (1.5-2.6) | 1.7 (1.1-2.3) | <0.001 |
| Monocytes(10$^9$/L), Median (IQR) | 0.2 (0.1-0.3) | 0.4 (0.3-0.5) | 0.6 (0.5-0.7) | 0.9 (0.7-1.1) | <0.001 |
| ALT(U/L), Median (IQR) | 20.0 (14.0-30.0) | 20.0 (15.0-29.0) | 21.0 (15.0-32.0) | 22.0 (14.0-36.0) | 0.121 |
| AST(U/L), Median (IQR) | 25.0 (19.0-35.0) | 24.0 (19.0-35.0) | 25.0 (19.0-37.0) | 27.0 (20.0-47.0) | <0.001 |
| GLU(U/L), Median (IQR) | 115.0 (100.0-136.0) | 117.0 (102.0-137.0) | 118.0 (102.0-139.0) | 119.0 (103.0-147.0) | 0.010 |
| SCR (mg/dl), Median (IQR) | 0.8 (0.7-1.1) | 0.9 (0.8-1.1) | 0.9 (0.8-1.1) | 1.0 (0.8-1.3) | <0.001 |
| BUN (mg/dl), Median (IQR) | 16.0 (13.0-21.0) | 16.0 (13.0-21.0) | 17.0 (14.0-21.0) | 18.0 (14.0-26.0) | <0.001 |
| EGFR (ml/min), Median (IQR) | 82.3 (64.0-100.0) | 81.8 (63.6-98.0) | 80.9 (62.4-97.0) | 73.5 (54.2-95.0) | <0.001 |
| SIRI, Median (IQR) | 0.7 (0.4-0.9) | 1.6 (1.3-1.8) | 2.9 (2.5-3.4) | 6.1 (4.8-8.8) | <0.001 |
| Hypertension, n(%) | 446 (61.9%) | 422 (58.5%) | 410 (56.9%) | 361 (50.1%) | <0.001 |
| Diabetes n(%) | 287 (39.8%) | 270 (37.5%) | 242 (33.6%) | 239 (33.2%) | 0.022 |
| Cerebrovascular disease, n(%) | 63 (8.7%) | 65 (9.0%) | 53 (7.4%) | 81 (11.2%) | 0.08 |
| Stroke, n(%) | 58 (8.0%) | 73 (10.1%) | 73 (10.1%) | 84 (11.7%) | 0.153 |
| Heart failure, n(%) | 58 (8.0%) | 73 (10.1%) | 73 (10.1%) | 84 (11.7%) | <0.001 |
| CHD, n(%) | 58 (8.0%) | 73 (10.1%) | 73 (10.1%) | 84 (11.7%) | 0.382 |
| Myocardial Infarction, n(%) | 203 (28.2%) | 217 (30.1%) | 213 (29.5%) | 241 (33.4%) | 0.164 |
| Shock, n(%) | 13 (1.8%) | 35 (4.9%) | 41 (5.7%) | 109 (15.1%) | <0.001 |
| Respiratory failure, n(%) | 35 (4.9%) | 40 (5.6%) | 55 (7.6%) | 124 (17.2%) | <0.001 |
| Surgery type, n(%) | | | | | 0.687 |
| Coronary artery bypass grafting, n(%) | 450 (62.4%) | 462 (64.1%) | 457 (63.4%) | 441 (61.2%) | |
| Others,n(%) | 271 (37.6%) | 259 (35.9%) | 264 (36.6%) | 280 (38.8%) | |
| AKI, n(%) | 535 (74.2%) | 549 (76.1%) | 566 (78.5%) | 578 (80.2%) | 0.038 |

(Q4) had a higher median BMI (28.94 kg/m$^2$ vs 27.33 kg/m$^2$, P<0.001). Q4 patients exhibited higher median heart rate (HR) (80 beats per minutes [bpm] vs 80 bpm, P<0.001), respiratory rate (RR) (16 respirations per minute [rpm] vs 16 rpm, P<0.001), and body temperature (36.61°C vs 36.44°C, P<0.001) compared to Q1. Hematological parameters showed significant differences between Q4 and Q1. Q4 patients had higher median white blood cell (WBC) count (11.9×10$^9$/L vs 7.6×10$^9$/L, P<0.001), neutrophil count (13.3×10$^9$/L vs 5.97×10$^9$/L, P<0.001), monocyte count (0.87×10$^9$/L vs 0.22×10$^9$/L, P<0.001), and PLT (180×10$^9$/L vs 152×10$^9$/L, P<0.001), while lymphocyte count was significantly lower (1.67×10$^9$/L vs 2.03×10$^9$/L, P<0.001). Renal function indicators also differed significantly. Q4 patients showed higher levels of Scr (1.0 mg/dL vs 0.8 mg/dL, P<0.001) and blood urea nitrogen (BUN) (18 mg/dL vs 16 mg/dL, P<0.001), with lower eGFR (73.45 mL/min/1.73m$^2$ vs 82.26 mL/min/1.73m$^2$, P<0.001) compared to Q1. Patients in higher SIRI quartiles were more likely to have shock (15.12% vs 1.80%, P<0.001), respiratory failure (17.20% vs 4.85%, P<0.001), and heart failure (11.65% vs 8.04%, P<0.001). Interestingly, the prevalence of hypertension (50.07% vs 61.86%, P<0.001) and diabetes (33.15% vs 39.81%, P=0.022) was lower in the higher SIRI quartiles. The distribution of surgery type was similar across SIRI quartiles, with CABG accounting for 62.4%, 64.1%, 63.4%, and 61.2% in Q1 through Q4, respectively (P=0.687).

### 3.2  Relationship between SIRI and severity of AKI

Fig 2 illustrated the relationship between SIRI levels and the severity of AKI. The results demonstrated a significant upward trend in SIRI levels corresponding to increasing AKI severity. The median SIRI values (with IQR) for each AKI

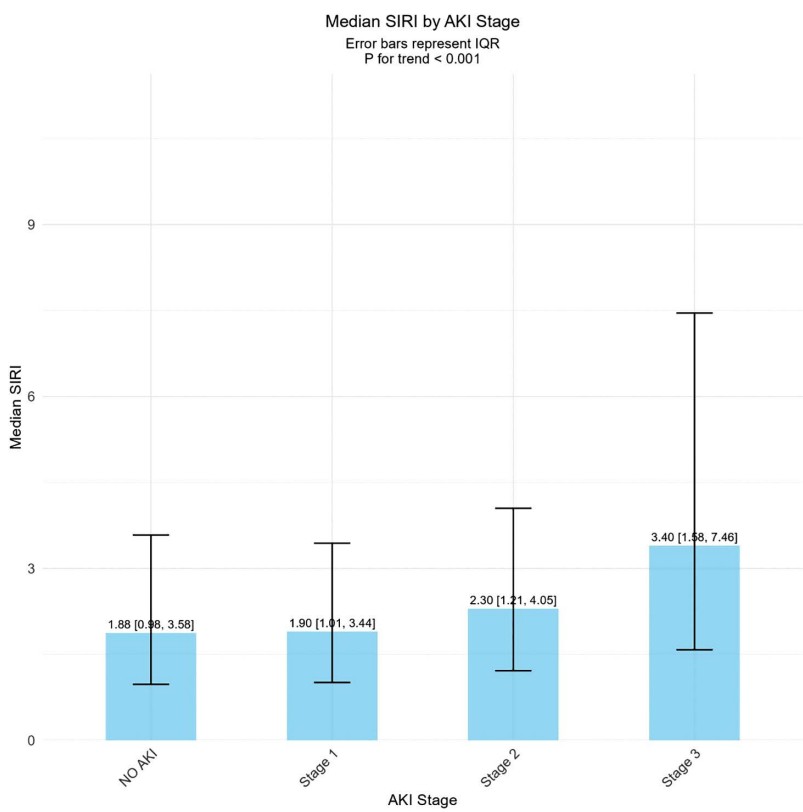

**Fig 2.  Distribution of SIRI Levels across Different Stages of AKI Severity.**

stage were as follows: No AKI: 1.88 (IQR: 0.98–3.58), AKI Stage 1: 1.90 (IQR: 1.01–3.44), AKI Stage 2: 2.30 (IQR: 1.21–4.05), and AKI Stage 3: 3.40 (IQR: 1.58–7.46). This ascending trend was statistically significant (P for trend < 0.001). The analysis revealed a strong association between SIRI levels and both the occurrence and severity of AKI. As the severity of AKI increased, patients exhibited correspondingly higher SIRI levels.

### 3.3 Association between SIRI and risk of AKI

Four logistic regression models were used to examine the association between SIRI and AKI occurrence, as shown in Table 2. Model 1 was unadjusted; Model 2 adjusted for age, sex, and race; Model 3 further adjusted for Hb, PLT, Scr, and eGFR; Model 4 additionally adjusted for history of hypertension, diabetes, heart failure, use of ACEIs or ARBs, use of loop diuretics, and surgery type. In the unadjusted Model 1, using Q1 as reference, we observed ORs for Q2, Q3, and Q4 of 1.11 (95% CI: 0.87–1.41), 1.27 (95% CI: 1.00–1.62), and 1.41 (95% CI: 1.10–1.80), respectively, indicating a positive correlation between SIRI and AKI risk. In Model 2, after adjusting for age, sex, and race, SIRI remained positively associated with AKI risk, albeit slightly attenuated, with ORs for Q2, Q3, and Q4 of 1.10 (95% CI: 0.86–1.40), 1.24 (95% CI: 0.96–1.57), and 1.40 (95% CI: 1.09–1.80), respectively. In the fully adjusted Model 4, the ORs were 1.11 (95% CI: 0.87–1.42), 1.25 (95% CI: 0.97–1.62), and 1.35 (95% CI: 1.04–1.77), respectively. When SIRI was treated as a continuous variable, a one-unit increase in SIRI was associated with a 2% (OR: 1.02, 95% CI: 1.00–1.04), 2% (OR: 1.02, 95% CI: 1.00–1.04), 1% (OR: 1.01, 95% CI: 1.00–1.03), and 2% (OR: 1.02, 95% CI: 1.00–1.04) increase in AKI risk in Models 1, 2, 3, and 4, respectively. Notably, a significant dose-response relationship between SIRI and AKI risk was observed across all models. P for trend values in Models 1, 2, 3, and 4 were 0.004, 0.006, 0.028, and 0.016, respectively, indicating that as SIRI levels increased, the risk of AKI also increased correspondingly. These findings indicated that elevated SIRI levels were associated with an increased risk of AKI across all models, with the relationship persisting after adjustment for multiple potential confounders. The association appeared most pronounced when comparing the highest and lowest SIRI quartiles.

### 3.4 Non-linear relationship analysis

Fig 3 illustrated the relationship between SIRI and AKI risk after adjusting for all covariates in Model 4. The curve demonstrated an approximately linear positive correlation between SIRI and AKI risk. As SIRI values increased, the probability of AKI occurrence showed a consistent upward trend. The P-value for the non-linearity test was 0.752, indicating no significant non-linear relationship between SIRI and AKI risk.

**Table 2. Association between SIRI and risk of AKI after cardiac surgery: Results from logistic regression models.**

| Variables | Model 1 | | Model 2 | | Model 3 | | Model 4 | |
|---|---|---|---|---|---|---|---|---|
| | OR (95%CI) | P value | OR (95%CI) | P value | OR (95%CI) | P value | OR (95%CI) | P value |
| Continuous | 1.02 (1.00, 1.04) | 0.062 | 1.02 (1.00, 1.04) | 0.041 | 1.01 (1.00, 1.03) | 0.178 | 1.02 (1.00, 1.04) | 0.080 |
| Q1 | — | | — | | — | | — | |
| Q2 | 1.11 (0.87, 1.41) | 0.394 | 1.10 (0.86, 1.40) | 0.434 | 1.10 (0.86, 1.41) | 0.429 | 1.11 (0.87, 1.42) | 0.414 |
| Q3 | 1.27 (1.00, 1.62) | 0.055 | 1.24 (0.96, 1.57) | 0.099 | 1.22 (0.95, 1.57) | 0.120 | 1.25 (0.97, 1.62) | 0.082 |
| Q4 | 1.41 (1.10, 1.80) | 0.007 | 1.40 (1.09, 1.80) | 0.010 | 1.31 (1.01, 1.71) | 0.040 | 1.35 (1.04, 1.77) | 0.026 |
| P For Trend | 1.12 (1.04, 1.21) | 0.004 | 1.12 (1.03, 1.21) | 0.006 | 1.10 (1.01, 1.19) | 0.028 | 1.11 (1.02, 1.21) | 0.016 |

Model 1: Unadjusted;

Model 2: Adjusted for age, sex, and race;

Model 3: Adjusted for variables in Model 1 + Hb, PLT, Scr, and eGFR;

Model 4: Adjusted for variables in Model 2 + history of hypertension, diabetes, heart failure, use of ACEI/ARB, and use of loop diuretics

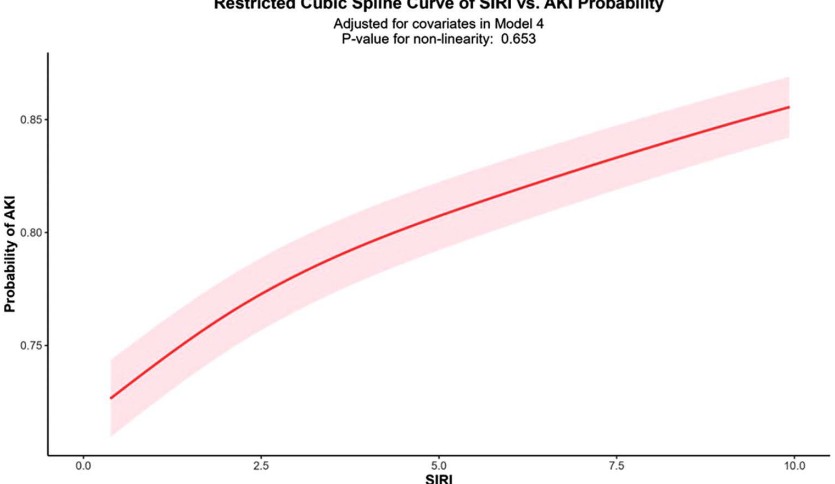

**Fig 3. RCS Analysis of the Association between SIRI and Risk of AKI.** Adjusted for all covariates in Model 4, including age, sex, race, Hb, PLT, Scr, eGFR, history of hypertension, diabetes, heart failure, use of ACEI/ARB, and use of loop diuretics.

## 3.5 Subgroup analysis and interaction effects

To assess the consistency of the association between SIRI and AKI risk across different populations, we conducted extensive subgroup analyses and interaction tests after adjusting for all covariates in Model 4. Table 3 summarized these results. Overall, the positive correlation between SIRI and AKI risk remained consistent across most subgroups. No statistically significant differences in the association between SIRI and AKI risk were observed across subgroups of sex, age, race, hypertension, diabetes, heart failure, stroke, CHD, and surgery type (all interaction P-values > 0.05). This suggested that SIRI's role as a risk factor for AKI was relatively stable across these populations. However, significant interactions were found between SIRI and history of myocardial infarction, respiratory failure, and loop diuretic use (P for interaction 0.040, 0.018, and 0.020, respectively). Specifically, the association between SIRI and AKI risk was more pronounced in patients with a history of myocardial infarction, without respiratory failure, and not using loop diuretics, with corresponding ORs of 1.261 (95% CI: 1.084–1.467), 1.113 (95% CI: 1.021–1.214), and 2.306 (95% CI: 1.200–4.434), respectively. Notably, the association between SIRI and AKI risk was observed in both CABG and non-CABG patients, with a numerically stronger effect in the non-CABG group (OR: 1.195, 95% CI: 1.044–1.368) compared to the CABG group (OR: 1.064, 95% CI: 0.959–1.180), although the interaction was not statistically significant (P for interaction = 0.171). These findings revealed potential variations in the association between SIRI and AKI risk among certain specific patient subgroups.

## 4. Discussion

The present study investigated the association between SIRI levels and post-cardiac surgery AKI in 2,884 patients undergoing cardiac surgery. We found a significant positive correlation between SIRI levels and both the incidence and severity of AKI. Furthermore, an approximately linear relationship between SIRI and AKI risk was observed. These findings not only emphasized the potential importance of inflammatory responses in the pathogenesis of AKI but also provided a new perspective for identifying high-risk patients in clinical practice.

AKI following cardiac surgery remains a major challenge in clinical practice that urgently needs to be addressed. This complication is not only characterized by its high incidence but, more importantly, by its potential to trigger a series of serious cascading effects. In the short term, AKI may impede patient recovery and prolong hospital stays. In the long term, it can significantly increase healthcare resource consumption and even pose a threat to patient survival [3,4].

**Table 3. Subgroup Analysis of the Association between SIRI and AKI Risk.**

| Variables | SIRI quartiles | | |
|---|---|---|---|
| | OR 95%CI | P value | P for interaction |
| Sex | | | 0.422 |
| Male | 1.086 (0.987, 1.195) | 0.093 | |
| Female | 1.173 (0.994, 1.384) | 0.059 | |
| Age | | | 0.334 |
| <60 years | 1.035 (0.887, 1.208) | 0.663 | |
| ≥60 years | 1.134 (1.026, 1.253) | 0.014 | |
| Race | | | 0.145 |
| White | 1.084 (0.981, 1.199) | 0.115 | |
| Black/Asia/Hispanic | 0.930 (0.720, 1.201) | 0.577 | |
| Other | 1.274 (1.045, 1.555) | 0.017 | |
| Hypertension | | | 0.530 |
| No | 1.074 (0.947, 1.218) | 0.269 | |
| Yes | 1.132 (1.015, 1.263) | 0.026 | |
| Diabetes | | | 0.980 |
| No | 1.106 (0.999, 1.224) | 0.052 | |
| Yes | 1.108 (0.963, 1.276) | 0.153 | |
| Heart Failure | | | 0.129 |
| No | 1.145 (1.041, 1.259) | 0.005 | |
| Yes | 0.976 (0.813, 1.173) | 0.799 | |
| Stroke | | | 0.847 |
| No | 1.104 (1.011, 1.205) | 0.028 | |
| Yes | 1.071 (0.804, 1.428) | 0.637 | |
| CHD | | | 0.629 |
| No | 1.059 (0.873, 1.284) | 0.561 | |
| Yes | 1.116 (1.017, 1.226) | 0.021 | |
| Myocardial Infarction | | | **0.040** |
| No | 1.046 (0.947, 1.155) | 0.378 | |
| Yes | 1.261 (1.084, 1.467) | 0.003 | |
| Shock | | | 0.070 |
| No | 1.104 (1.013, 1.202) | 0.024 | |
| Yes | 0.648 (0.349, 1.202) | 0.169 | |
| Respiratory Failure | | | **0.018** |
| No | 1.113 (1.021, 1.214) | 0.015 | |
| Yes | 0.674 (0.436, 1.042) | 0.076 | |
| ACEI/ARB | | | 0.2932 |
| No | 1.073 (0.971, 1.185) | 0.166 | |
| Yes | 1.180 (1.015, 1.371) | 0.031 | |
| Loop Diuretics | | | **0.020** |
| No | 2.306 (1.200, 4.434) | 0.012 | |
| Yes | 1.088 (1.000, 1.185) | 0.050 | |
| CABG | | | 0.171 |
| No | 1.195 (1.044, 1.368) | 0.012 | |
| Yes | 1.064 (0.959, 1.180) | 0.243 | |

Adjusted for all covariates in Model 4, including age, sex, race, Hb, PLT, Scr, eGFR, history of hypertension, diabetes, heart failure, use of ACEI/ARB, and use of loop diuretics.

Therefore, identifying effective predictive tools is crucial for improving patient prognosis. Previous studies have explored the associations between various inflammatory markers and post-cardiac surgery AKI. Rossi et al., in a study of 2,034 acute coronary syndrome patients undergoing PCI, found that high-sensitivity CRP was associated with an increased risk of AKI [8]. A prospective study by Lakhal et al. on 82 cardiac surgery patients demonstrated that interleukin-6 showed good performance in predicting moderate to severe AKI in cardiac surgery patients [16]. Chen et al.'s research on 402 patients revealed that a cytokine model composed of interferon-γ, interleukin-16, and MIP-1α performed excellently in predicting severe AKI [9]. Additionally, another study by Chen et al., analyzing 306 cardiac surgery patients, identified interferon-γ and stem cell growth factor-β as potential novel predictive biomarkers for post-cardiac surgery AKI [17]. More recently, composite inflammatory indices have attracted increasing attention. The systemic immune-inflammation index (SII) has been shown to be associated with AKI after cardiac surgery in a large cohort study using the MIMIC-IV database [18]. SIRI has also demonstrated predictive value for AKI in other clinical settings, including acute pancreatitis [19] and contrast-associated AKI following elective PCI [20]. However, the specific relationship between SIRI and cardiac surgery-associated AKI has not been investigated. The present study, by investigating the association between SIRI and post-cardiac surgery AKI, not only provided a potential new predictive tool but also offered a novel perspective for understanding the comprehensive role of inflammation in AKI occurrence, filling a gap in existing research.

The association between SIRI, an indicator comprehensively reflecting inflammatory status, and post-cardiac surgery AKI can potentially be explained through the complex interactions of neutrophils, monocytes, and lymphocytes. These three cell types play crucial roles in the onset and progression of AKI, and SIRI is calculated based on their counts. Ischemia-reperfusion injury during cardiac surgery is a significant factor leading to AKI, with neutrophil recruitment and activation playing a vital role in renal damage. Activated neutrophils directly cause tubular epithelial cell injury by releasing neutrophil extracellular traps (NETs), reactive oxygen species (ROS), and proteases [21]. Concurrently, elevated levels of monocytes, particularly intermediate monocytes (CD14++CD16+), are closely associated with increased AKI risk following cardiac surgery [22]. Upregulated expression of TLR4, TLR2, and CD64 on these cells may activate inflammatory signaling pathways, promoting the release of inflammatory factors and exacerbating renal damage [23]. Moreover, post-cardiac surgery immune dysregulation plays a significant role in AKI occurrence. There exists a complex bidirectional interaction between AKI and the immune system, with both innate and adaptive immune responses participating in renal injury and recovery processes [24]. This process involves the joint participation of non-classical monocytes and lymphocytes [25]. Non-classical monocytes may aggravate renal damage by promoting inflammatory responses, while a decrease in lymphocyte count may lead to reduced immunoregulatory function, failing to effectively suppress excessive inflammatory responses, thereby increasing AKI risk [26]. In our study, we observed a significant association between SIRI and the risk of AKI in post-cardiac surgery patients, potentially reflecting the comprehensive role of these cell-mediated complex inflammatory and immune responses in AKI occurrence. The unique advantage of SIRI lies in its ability to simultaneously capture changes in these three cell types, thus providing a more comprehensive and accurate assessment of inflammatory status. However, the specific molecular pathways and precise mechanisms of cellular interactions of SIRI components in AKI occurrence still require further in-depth research for elucidation.

This study was the first to comprehensively investigate the association between SIRI and the risk of AKI following cardiac surgery. Our findings not only introduced a novel, integrative inflammatory indicator associated with post-cardiac surgery AKI but also offered valuable insights into its clinical application and limitations. Our results demonstrated that SIRI, a simple and easily obtainable metric, showed a modest but statistically significant association with AKI risk in cardiac surgery patients. Unlike previous studies that primarily focused on single inflammatory markers [27], SIRI integrated information from neutrophils, monocytes, and lymphocytes, offering a more comprehensive assessment of inflammatory status. This holistic approach provided a more nuanced understanding of the inflammatory milieu associated with post-cardiac surgery AKI.

Importantly, our study revealed that SIRI was not only associated with the occurrence of AKI but also positively correlated with its severity. This finding expanded the utility of existing AKI risk assessment models, potentially allowing for more precise stratification of patients based on their risk for severe AKI. The RCS analysis further revealed an approximately linear relationship between SIRI and AKI risk, implying a continuous, stable correspondence between increasing SIRI values and rising AKI risk. This linear relationship may provide clinicians with a supplementary risk stratification marker, offering a refined basis for individualized risk assessment and the formulation of precise prevention strategies.

The unique advantage of SIRI lay in its simplicity, cost-effectiveness, and ease of acquisition. Requiring only routine blood tests for calculation without additional testing costs, SIRI may have application potential in clinical practice. This accessibility could facilitate its evaluation in various healthcare settings, although further validation is needed before clinical implementation.

Our subgroup analysis revealed the complexity of the association between SIRI and AKI risk in specific patient populations. The association remained consistent across most subgroups, including sex, age, race, and surgery type. However, significant interactions were observed between SIRI and history of myocardial infarction, respiratory failure, and loop diuretic use. The stronger association between SIRI and AKI risk in patients with a history of myocardial infarction may be attributed to the chronic inflammatory state often present in these patients [28]. This heightened inflammatory background could potentially sensitize these patients to surgery-induced inflammatory responses, exacerbating their risk for AKI. Interestingly, we observed a weakened association between SIRI and AKI risk in patients with respiratory failure. This seemingly paradoxical result may reflect the complex pathophysiological state of these patients. Previous studies have suggested a potential protective role of HIF activation for the kidneys in ischemic AKI [29], which might partially explain this observation. The interaction between diuretic use and SIRI revealed potential impacts of pharmacological interventions on the inflammation-AKI axis. The more significant association between SIRI and AKI risk in patients not using diuretics could be due to the partial alleviation of inflammation-mediated renal injury by diuretics through improved renal hemodynamics [30]. However, the use of diuretics in AKI management remains controversial, with some studies cautioning against potential harm [31].

In conclusion, this study demonstrated a modest but significant association between SIRI and AKI following cardiac surgery. Its integration of multiple inflammatory cell types provided a comprehensive assessment of inflammatory status, which may contribute to risk stratification for post-cardiac surgery AKI. The varying strength of association across different patient subgroups suggested the need for individualized risk assessment strategies. However, given the modest effect size, further research is warranted to validate these findings in external cohorts, explore the underlying mechanisms, and determine whether SIRI-guided interventions could help reduce the incidence and severity of post-cardiac surgery AKI.

## 5. Limitations

While our study provided compelling evidence for the association between SIRI and post-cardiac surgery AKI, several limitations should be acknowledged. First, the retrospective nature of our study may have introduced potential biases. Second, while SIRI offered a comprehensive view of inflammatory status, it did not capture all aspects of the complex inflammatory response following cardiac surgery. Third, due to the limitations of the MIMIC database, detailed intraoperative variables such as cardiopulmonary bypass (CPB) use, CPB duration, aortic cross-clamp time, and blood transfusion were not available for analysis. These factors are known to influence postoperative AKI risk and should be considered in future prospective studies. Fourth, the study did not assess the long-term outcomes associated with SIRI and AKI. Finally, while our findings were promising, external validation in prospective studies is necessary to confirm the association between SIRI and post-cardiac surgery AKI.

## 6. Conclusion

This study demonstrated that SIRI was modestly but significantly associated with AKI following cardiac surgery. SIRI's association with AKI risk and severity persisted after adjusting for multiple confounders, highlighting its potential as a

supplementary risk stratification marker. The varying strength of this association across different patient subgroups underscored the importance of individualized risk assessment. As a readily available and cost-effective measure, SIRI may provide additional value to existing AKI risk assessment, although external validation is needed before clinical application.

## Supporting Information

**Supplementary Material S1.**
(DOCX)

## Author contributions

**Conceptualization:** Huiliang Xie, Keyang Zheng.

**Data curation:** Huiliang Xie, Keyang Zheng.

**Formal analysis:** Huiliang Xie, Keyang Zheng.

**Investigation:** Kun Wang.

**Methodology:** Kun Wang.

**Project administration:** Xiayang Zhu.

**Resources:** Xiayang Zhu.

**Supervision:** Huiliang Xie, Keyang Zheng, Xiayang Zhu.

**Validation:** Zixu Zhao, Jingyi Xiao, Yuzhe Yin.

**Visualization:** Rite Si.

**Writing – original draft:** Huiliang Xie, Keyang Zheng.

**Writing – review & editing:** Huiliang Xie, Keyang Zheng, Kun Wang, Zixu Zhao, Rite Si, Jingyi Xiao, Yuzhe Yin, Xiayang Zhu.

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
