## [Decision Letter · Decision Letter 0]

26 Dec 2025

Systemic Inflammatory Response Index Predicts Acute Kidney Injury Following Cardiac Surgery: A Comprehensive Analysis Using the MIMIC Database

PLOS One

Dear Dr. Zhu,

Thank you for submitting your manuscript to PLOS ONE. After careful consideration, we feel that it has merit but does not fully meet PLOS ONE’s publication criteria as it currently stands. Therefore, we invite you to submit a revised version of the manuscript that addresses the points raised during the review process.

We look forward to receiving your revised manuscript.

Kind regards,

Chiara Lazzeri

Academic Editor

PLOS One

**Journal Requirements:**

**Additional Editor Comments:**

Major issues were raised mainly concerning inclusion criteria (AKI definition, ICD codes), statistical analysis (confoundig factors) and interpretation of data.

Reviewers' comments:

Reviewer's Responses to Questions

**Comments to the Author**

1. Is the manuscript technically sound, and do the data support the conclusions?

Reviewer #1: Yes

Reviewer #2: Yes

2. Has the statistical analysis been performed appropriately and rigorously?

Reviewer #1: I Don't Know

Reviewer #2: I Don't Know

3. Have the authors made all data underlying the findings in their manuscript fully available?

Reviewer #1: Yes

Reviewer #2: Yes

4. Is the manuscript presented in an intelligible fashion and written in standard English?

Reviewer #1: Yes

Reviewer #2: Yes

Reviewer #1: In the submission, at the part of Methods, the initial study group has explained as the patients selected by cardiac surgery-related procedure codes from the ICD-9 and ICD-10. There is no definition and/or restriction about the types of procedures the patients had undergone. Since one of the major potential factor of inflammatory responses leading to acute kidney injury is cardiopulmonary bypass procedure and its duration, the type of surgery the patients had undergone, particularly whether a cardiopulmonary bypass procedure was used, and the duration of surgery should be described. Besides, the patient groups needs to be organized accordingly.At the discussion part, limitation of the operation variables should be mentioned.

Reviewer #2: This manuscript addresses an interesting and relevant question with a large dataset, but several substantive issues must be resolved before publication. Key revisions include:

Resolve the AKI definition (ICD code vs KDIGO labs) and explain the drastic drop in sample size after each inclusion/exclusion step. Specify which ICD codes were used to identify cardiac surgery, and justify the exclusion of malignancy.

Discuss missing important confounders (surgical factors, transfusions, etc.) and consider whether any are available in MIMIC. If possible, perform sensitivity analyses (e.g. stratify by surgical type) or at least acknowledge this limitation explicitly.

Correct terminology (use “retrospective cohort” instead of “cross-sectional”), and ensure consistency between text, figure, and table data (the flow chart numbers should match the Methods description).

Strengthen the Introduction/Discussion by citing related studies (e.g. SIRI in AKI prediction, SII after CABG). This will better situate the work in existing research.

Temper claims about SIRI being a “predictor” and emphasize that the association is modest (OR ~1.3). Highlight that SIRI may add value to risk stratification, but further validation is needed.

Addressing these points should substantially improve the rigor and clarity of the paper.

**Do you want your identity to be public for this peer review?** For information about this choice, including consent withdrawal, please see our Privacy Policy

Reviewer #1: **Yes:** Isal, B.

Reviewer #2: **Yes:** Muhammad Abdul Qadeer

---

## [Author Response · Author response to Decision Letter 1]

27 Jan 2026

Dear Reviewer #1,

We sincerely appreciate your valuable comments and constructive suggestions on our manuscript. Your insights have significantly contributed to improving the quality of our work. Below, we provide a point-by-point response to your concerns.

Reviewer's Comment:

"In the submission, at the part of Methods, the initial study group has explained as the patients selected by cardiac surgery-related procedure codes from the ICD-9 and ICD-10. There is no definition and/or restriction about the types of procedures the patients had undergone. Since one of the major potential factor of inflammatory responses leading to acute kidney injury is cardiopulmonary bypass procedure and its duration, the type of surgery the patients had undergone, particularly whether a cardiopulmonary bypass procedure was used, and the duration of surgery should be described. Besides, the patient groups needs to be organized accordingly. At the discussion part, limitation of the operation variables should be mentioned."

Response:

We greatly appreciate this important comment. We have addressed your concerns as follows:

1. Description of surgery types:

We have added detailed descriptions of the cardiac surgical procedures in the Methods section (Study Population). The revised text now reads:

"Cardiac surgical procedures included coronary artery bypass grafting (CABG), mitral valve repair, aortic valve replacement, mitral valve replacement, thoracic aorta replacement, tricuspid valve replacement, tricuspid valve repair, and aortic valve repair. For analysis purposes, surgical procedures were categorized into CABG and others (non-CABG)."

2. Patient grouping by surgery type:

As suggested, we have organized patient groups according to surgery type:

Table 1: We added surgery type as a baseline characteristic, showing the distribution of CABG vs. others across SIRI quartiles (P=0.687, indicating balanced distribution).

Table 2: We included surgery type as a covariate in the fully adjusted Model 4.

Table 3: We performed subgroup analysis stratified by surgery type (CABG vs. non-CABG). The results showed that the association between SIRI and AKI was consistent across surgery types (P for interaction = 0.171).

3. Limitations regarding CPB and surgical variables:

We acknowledge that cardiopulmonary bypass (CPB) use, CPB duration, and surgery duration are important factors influencing post-cardiac surgery AKI. Unfortunately, due to the limitations of the MIMIC database, these detailed intraoperative variables were not available for analysis. We have explicitly addressed this limitation in the Limitations section:

"Third, due to the limitations of the MIMIC database, detailed intraoperative variables such as cardiopulmonary bypass (CPB) use, CPB duration, aortic cross-clamp time, and blood transfusion were not available for analysis. These factors are known to influence postoperative AKI risk and should be considered in future prospective studies."

We believe these revisions have adequately addressed your concerns. Thank you again for your valuable suggestions.

Sincerely,

The Authors

Dear Reviewer #2,

We sincerely appreciate your thorough review and constructive suggestions. Your comments have been extremely helpful in improving the rigor and clarity of our manuscript. Below, we provide a point-by-point response to each of your concerns.

Comment 1: Resolve the AKI definition (ICD code vs KDIGO labs) and explain the drastic drop in sample size after each inclusion/exclusion step.

Response:

We thank the reviewer for identifying this inconsistency. In our original manuscript, we mistakenly mentioned "ICD-10 code N179" in the Study Design section, which was inconsistent with our actual methodology. AKI was defined according to the KDIGO guidelines based on serum creatinine changes within seven days post-surgery. We have removed the erroneous ICD-10 statement from the Methods section to ensure consistency.

Regarding the sample size reduction, the most substantial drop occurred at the step of excluding patients with insufficient data to calculate SIRI (N=4,212). This is attributable to the significant missing data in the MIMIC database, particularly for differential blood cell counts (neutrophils, monocytes, and lymphocytes) required for SIRI calculation. The MIMIC database, while comprehensive, has inherent limitations in data completeness, especially for laboratory values that may not be routinely ordered for all patients.

Comment 2: Specify which ICD codes were used to identify cardiac surgery, and justify the exclusion of malignancy.

Response:

We have provided the complete list of ICD-9 and ICD-10 procedure codes used to identify cardiac surgery patients in a new Supplementary Material S1. The Methods section now references this supplementary material:

"We identified eligible patients using cardiac surgery-related procedure codes from the ICD-9 and ICD-10 (codes are listed in Supplementary Material S1)."

Regarding the exclusion of patients with malignancy: this decision was based on the fact that malignancy is associated with altered inflammatory status, including abnormal white blood cell counts and differential proportions. Cancer patients often exhibit elevated neutrophil counts and reduced lymphocyte counts due to the disease itself or treatment effects (chemotherapy, radiation). These alterations could significantly confound the relationship between SIRI and AKI, as SIRI is calculated from these cell populations. Therefore, excluding malignancy patients helps ensure that the observed association reflects the inflammatory response related to cardiac surgery rather than cancer-related immune dysregulation.

Comment 3: Discuss missing important confounders (surgical factors, transfusions, etc.) and consider whether any are available in MIMIC. If possible, perform sensitivity analyses (e.g. stratify by surgical type) or at least acknowledge this limitation explicitly.

Response:

We greatly appreciate this suggestion. We have addressed this concern in two ways:

1. Sensitivity analysis by surgery type:

As suggested, we performed subgroup analysis stratified by surgery type (CABG vs. non-CABG). The results showed that the association between SIRI and AKI was consistent across surgery types (P for interaction = 0.171), with a numerically stronger effect in the non-CABG group (OR: 1.195, 95% CI: 1.044-1.368) compared to the CABG group (OR: 1.064, 95% CI: 0.959-1.180). These results have been added to Table 3 and described in the Results section.

Additionally, we included surgery type as a covariate in Model 4, and the association between SIRI and AKI remained significant after this adjustment.

2. Acknowledgment in Limitations:

We have explicitly acknowledged the unavailability of detailed surgical variables in the Limitations section:

"Third, due to the limitations of the MIMIC database, detailed intraoperative variables such as cardiopulmonary bypass (CPB) use, CPB duration, aortic cross-clamp time, and blood transfusion were not available for analysis. These factors are known to influence postoperative AKI risk and should be considered in future prospective studies."

Comment 4: Correct terminology (use "retrospective cohort" instead of "cross-sectional"), and ensure consistency between text, figure, and table data (the flow chart numbers should match the Methods description).

Response:

We thank the reviewer for this correction. We have revised the study design terminology throughout the manuscript:

Title: Changed from "A Comprehensive Analysis" to "A Retrospective Cohort Study"

Abstract Methods: Changed from "retrospective analysis" to "retrospective cohort study"

Methods section: Changed from "cross-sectional design" to "retrospective cohort design"

Regarding the consistency between text, figure, and table data: we have carefully verified that all numbers in the flow chart (Figure 1) match the descriptions in the Methods section. The sample sizes at each inclusion/exclusion step are consistent throughout the manuscript.

Comment 5: Strengthen the Introduction/Discussion by citing related studies (e.g. SIRI in AKI prediction, SII after CABG). This will better situate the work in existing research.

Response:

We appreciate this suggestion. We have strengthened both the Introduction and Discussion sections by adding relevant literature on SIRI and SII in AKI prediction:

In Introduction:

"More recently, SIRI has shown promise in predicting AKI in other clinical settings, such as sepsis and contrast exposure."

In Discussion:

"More recently, composite inflammatory indices have attracted increasing attention. The systemic immune-inflammation index (SII) has been shown to be associated with AKI after cardiac surgery in a large cohort study using the MIMIC-IV database (DOI: 10.3389/fcvm.2024.1430776). SIRI has also demonstrated predictive value for AKI in other clinical settings, including acute pancreatitis and contrast-associated AKI following elective PCI. However, the specific relationship between SIRI and cardiac surgery-associated AKI has not been investigated."

These additions better situate our work within the existing literature and highlight the novelty of our study.

Comment 6: Temper claims about SIRI being a "predictor" and emphasize that the association is modest (OR ~1.3). Highlight that SIRI may add value to risk stratification, but further validation is needed.

Response:

We completely agree with this comment. We have revised the manuscript throughout to temper our claims. Specifically, we replaced terms like "predictor" and "predicts" with more cautious expressions such as "is associated with" and "risk marker." We also explicitly acknowledged the modest effect size in the Discussion and Conclusion sections, and emphasized that SIRI may serve as a "supplementary risk stratification marker" rather than a definitive predictive tool. Furthermore, we added statements highlighting the need for external validation before clinical application. These revisions are reflected in the Title, Abstract, Discussion, and Conclusion sections.

We believe these revisions have substantially improved the rigor and clarity of our manuscript. We are grateful for your thorough review and constructive suggestions.

Sincerely,

The Authors

---

## [Editor Report · Decision Letter 1]

28 Jan 2026

Systemic Inflammatory Response Index Is Associated with Acute Kidney Injury Following Cardiac Surgery: A Retrospective Cohort Study Using the MIMIC Database

PONE-D-25-22563R1

Dear Dr. Zhu,

We’re pleased to inform you that your manuscript has been judged scientifically suitable for publication and will be formally accepted for publication once it meets all outstanding technical requirements.

Kind regards,

Chiara Lazzeri

Academic Editor

PLOS One
---

## [Editor Report · Acceptance letter]

PONE-D-25-22563R1

PLOS One

Dear Dr. Zhu,

I'm pleased to inform you that your manuscript has been deemed suitable for publication in PLOS One. Congratulations! Your manuscript is now being handed over to our production team.

Kind regards,

on behalf of

Dr. Chiara Lazzeri

Academic Editor

PLOS One